# Thinking beyond Vaccination: Promising Add-On Strategies to Active Immunization and Vaccination in Pandemics—A Mini-Review

**DOI:** 10.3390/v15061372

**Published:** 2023-06-14

**Authors:** Franz Tatzber, Willibald Wonisch, Ulrike Resch, Wolfgang Strohmaier, Meinrad Lindschinger, Sabrina Mörkl, Gerhard Cvirn

**Affiliations:** 1Omnignostica Ltd., 3421 Höflein an der Danube, Austria; franz@tatzber.at; 2Department of Vascular Biology and Thrombosis Research, Medical University of Vienna, 1090 Vienna, Austria; uresch@gmx.at; 3Dr. Strohmaier & Partner Ltd., 1050 Vienna, Austria; w.strohmaier@scipharm.eu; 4Outpatient Clinic Laßnitzhöhe, Institute of Nutritional and Metabolic Diseases, 8301 Laßnitzhöhe, Austria; meinrad@lindschinger.at; 5Division of Medical Psychology, Psychosomatics and Psychotherapeutic Medicine, Medical University of Graz, 8036 Graz, Austria; sabrina.moerkl@medunigraz.at; 6Otto Loewi Research Center, Division of Physiological Chemistry, Medical University of Graz, 8010 Graz, Austria; gerhard.cvirn@medunigraz.at

**Keywords:** hypertonic salt solution, impregnation, filtering face piece, human monoclonal antibody, dry blood spot

## Abstract

There is little doubt that final victories over pandemics, such as COVID-19, are attributed to herd immunity, either through post-disease convalescence or active immunization of a high percentage of the world’s population with vaccines, which demonstrate protection from infection and transmission and are available in large quantities at reasonable prices. However, it is assumable that humans with immune defects or immune suppression, e.g., as a consequence of allograft transplantation, cannot be immunized actively nor produce sufficient immune responses to prevent SARS-CoV-2 infections. These subjects desperately need other strategies, such as sophisticated protection measures and passive immunization. Hypertonic salt solutions attack vulnerable core areas of viruses; i.e., salt denatures surface proteins and thus prohibits virus penetration of somatic cells. It has to be ensured that somatic proteins are not affected by denaturation regarding this unspecific virus protection. Impregnating filtering facepieces with hypertonic salt solutions is a straightforward way to inactivate viruses and other potential pathogens. As a result of the contact of salt crystals on the filtering facepiece, these pathogens become denatured and inactivated almost quantitatively. Such a strategy could be easily applied to fight against the COVID-19 pandemic and other ones that may occur in the future. Another possible tool to fight the COVID-19 pandemic is passive immunization with antibodies against SARS-CoV-2, preferably from human origin. Such antibodies can be harvested from human patients’ sera who have successfully survived their SARS-CoV-2 infection. The disadvantage of a rapid decrease in the immunoglobulin titer after the infection ends can be overcome by immortalizing antibody-producing B cells via fusion with, e.g., mouse myeloma cells. The resulting monoclonal antibodies are then of human origin and available in, at least theoretically, unlimited amounts. Finally, dry blood spots are a valuable tool for surveilling a population’s immunity. The add-on strategies were selected as examples for immediate, medium and long-term assistance and therefore did not raise any claim to completeness.

## 1. Impact of Salt Treatment in Filtering Facepieces in the Time of the Pandemic

Significant achievements in humankind are related to the ability to prevent foodstuff from microbial spoilage. The early Sumerians, Babylonians, and the Jewish Kingdom of approximately 1600 BC already used salt as a dietary essential and for food preservation [1]. Neither America’s discovery nor the ultimate proof that our planet is bullet shaped would have been possible without food preservation. Primarily, meat and fish were easily devastated in a temperature-dependent way. The knowledge that these foods might gain durability through excessive salting was essential for many expeditions, including Darwin and Humboldt. One of the reasons for the improved food preservation of hypertonic saline is that sodium chloride causes peptide self-folding and the binding of chloride to hydrophobic cavities. This results in less surface hydrophobicity, which in turn reduces the bitter taste [2]. As salt’s influence on proteins led to their structural decay, salt-denaturation of proteins and nucleic acids became a rather feared event. This unspecific effect of high salt concentrations on peptides, proteins, and nucleic acids is common and leads to water loss and cleavage of hydrogen bonds, followed by aggregation and structural malformation in these molecules. Virus particles consist predominantly of nucleic acids and proteins and may be especially sensitive to denaturation by hypertonic saline because their surface proteins become destructured, and their binding capacity to cell receptors is strongly reduced if not inhibited. Although there are halophilic microorganisms that survive high salt concentrations, these microorganisms are specialized and are constantly dependent on high salt concentrations, so they have little chance of survival outside salt lakes or oceans. These halobacteria are killed under a salt concentration of less than 2.8 M, making them highly vulnerable to environmental conditions such as rain and wind [3]. Due to their safe and effective effects, hypertonic saline solutions are also used in medicine, where they are applied for various purposes. Among others, hypertonic sodium saline improves neurological effects in resuscitation for traumatic brain injury and has positive effects on increased intracranial pressure [4], as well as preserving anatomical specimens to prevent contamination [5,6] and improving host cell protein clearance in Protein A chromatography, the capture step for monoclonal antibody purification [7]. Finally, it should be mentioned that increased availability of NaCl can enhance antiviral activity against a broad spectrum of viral infections through an innate antiviral mechanism that produces hypochlorous acid when chloride ions are available [8].

Usually, microorganisms and viruses remain active, even if a face mask filters them. Estimations state that such masks retain approximately 70 per cent of the virus load if the mask is worn correctly [9,10]. We can assume that the majority of all pathogens present in breathed air will end up without significant loss of infectivity on the mask surfaces [11] if we apply these estimations to all pathogens available by air filtration. Therefore, after continuous wearing for several hours, which is common in clinical practice, it can be assumed that it will lead to a severe accumulation of pathogens, including SARS-CoV-2, on the surface and in the tissue of the filtering facepiece (FFP). Consequently, care is necessary by donning and doffing so that these microorganisms are not transferred by the wearer’s hands to mucous membranes, e.g., nose or eyes, to prohibit self-infection. However, donning and doffing of FFP are often associated with a great susceptibility to error among the general population. People often imitate the behavior of experts, health workers, or politicians in the media. However, a working group from France showed that, unfortunately, only 70% of healthcare workers used FFP according to the good practice guidelines in the photographs on leading information websites [12]. In this regard, a randomized control trial could also be classified, in which no significant difference in the risk of infection with SARS-CoV-2 could be shown between a control group (2.1%) compared to an FFP-wearing group (1.8%), which was endorsed by Jefferson et al. [13,14]. There should also be a focus on wearing FFP for infected people (source control), with a benefit at the population level [15]. Conversely, pathogens exhaled by contagious wearers of filtering facepieces may also remain in the facepiece tissue, leading to wearers’ contagion by mistake. Of course, such threats can easily be avoided by remembering the essential rules for hand and face hygiene. However, even accidental germ transfer to mucous membranes present in the eyes, nose, mouth, and lungs will lead to the contraction of a disease. These considerations are also important from the point of view of the environmental impact of using filtering facepieces, since longer wearing times and reprocessing have been extremely sparsely studied. Appropriate guidelines would also be required in this regard [16].

Furthermore, certain parts of the human population cannot be vaccinated against pathogens. Predominantly, people with transplantations get a mention in this context, but also, patients with severe allergies or autoimmune diseases, as well as subjects with an intolerance to ingredients of the vaccine, may be at high risk for vaccination. Finally, there will be persons who received vaccinations but did not react sufficiently to their active immunizations. All these people should be protected from infections by alternative methods, including the wearing of filtering facepieces, despite the risk of active pathogen accumulation, as described before. The impregnation of filtering facepieces with hypertonic salt solutions is a prevention measure to minimize the risk of self-contagion [11,17]. Nevertheless, an evaluation of the effective salt concentration is required for specific microorganisms, although a hypertonic salt solution of 10% sodium chloride was associated with significant inactivation in most microorganisms tested.

## 2. Salt Impregnation of Filtering Facepieces

Treatment of filtering facepieces with hypertonic salt solutions must be simple to apply safely by all humans, should not be restricted to people with laboratory experience, and should ideally already be carried out in the production process. First, it is necessary to obtain sufficient salt by preparing a hypertonic salt solution with ≧10% sodium chloride (NaCl). The FFP should consist of cloth material with good suction properties. The filtering facepiece should be soaked or sprinkled with the hypertonic sodium chloride solution. After coating, the FFP should be allowed to dry. Evaporation is achieved by sunlight, heat treatment with warm (room temperature) or hot air, and even ironing. None of these treatments will destroy salt or prevent its action. The dry FFP is then ready for use and should cover the mouth and nose of the wearer. Furthermore, care must be taken so that breathed air must pass through the mask and not pass by the filtering facepiece. If salt-impregnated FFPs are treated like that, extended quantitative and paramagnetic filtration and inactivation of pathogens are attainable; e.g., the infection rate for aerosolized alpha-coronavirus-1 in pigs (transmissible gastroenteritis virus—a surrogate for the SARS-CoV-2 virus) was only minimally reduced with untreated filtering facepieces. In contrast, the infection rate with salt-impregnated FFPs was significantly reduced by four orders of magnitude (10^4^), as indicated in Table 1, which was published previously [11].

As far and as soon as these pathogens become inactivated, they are no longer infectious, or at least their infectivity is reduced for the most part. While untreated masks can only protect the wearer’s surroundings, the person still depends on others’ discipline. An additional advantage of the hypertonic salt-impregnated FFP is the protection of the wearer’s surroundings and the wearer himself.

## 3. Mode of Action and Advantages of Filtering Facepieces with Salt Impregnation

The main property which improves the efficacy of FFPs is the hygroscopic nature of salt. It is generally accepted that germs need humid surroundings to develop their full pathogenicity. In this respect, it is allegeable that the vast majority of microorganisms and viruses are spread via aerosols and water droplets. Hygroscopic salt crystals preferentially attract these aerosols and water droplets in a paramagnetic way. The excellent solubility of sodium chloride (NaCl) in water leads to a considerable salt concentration in these droplets. In hypertonic conditions, the dissolved NaCl starts cleaving hydrogen bridge bonds responsible for the final structure of proteins. The structural integrity of proteins is lost by denaturation, and the virus surface proteins lose their ability to bind to receptor proteins on the surface of somatic cells, thus prohibiting the permeation of virus nucleic acids into the cells. Viruses cannot reproduce autonomously and cannot multiply without host cells. Therefore, virus replication is virtually impossible under such conditions, and infectivity and subsequent pathogenicity are broadly forfeited.

Additionally, those viruses and microorganisms sticking on masks’ surfaces are also inactivated by protein and nucleic acid denaturation and therefore hardly represent any risk for infection anymore. This also means protection to a large extent against infection by other people who refuse to wear FFPs. Salt impregnation of filtering facepieces inactivates germs [18], bacteria [19], and viruses [11,17] spread by others and may protect the wearers themselves. Harsh conditions, such as a humid environment that occurs with a prolonged period of use, even increase the inactivation activity [19]. Furthermore, as salt extends the shelf life of filtering facepieces due to the prohibited infectivity of microorganisms trapped and accumulated on the masks’ surfaces, a positive effect on waste management by salt impregnation of FFPs is assumable.

Of course, it is worth discussing using chemicals or salts other than NaCl for mask impregnation. Several salts are suited for impregnation, including potassium chloride [19], ammonium sulfate, and several nitrates [1]. Even salts of copper and silver can be applied on mask surfaces. However, sodium chloride is preferred mainly due to its nontoxic nature. Furthermore, inhalation of NaCl has beneficial effects on lung infections in general. On the other hand, small amounts of ethanol or isopropanol lead to quicker evaporation of liquid components. Finally, adding antiviral components, such as extracts from *Hedera* sp. or *Cistus incanus*, could further improve the efficacy of the antiviral activity of salt-impregnated filtering facepieces. Of course, it should be noted that the inactivation of microorganisms through protein and nucleic acid denaturation by salt lacks specificity. One significant advantage of the nonspecific nature of salt-impregnated FFPs is the generalized impact to be prepared against future pandemics of almost any origin by such simple personal security tools. However, salt impregnation should be avoided if hypersensitivity to salt is already present or occurs due to wearing an FFP coated with hypertonic saline.

## 4. Effect of Salt Solutions on Yeast, Viruses, and Airborne Microorganisms

Previously, some reports presented some experimental proof for these assumptions. Yeast cell growth was completely inhibited by a salt concentration of 10% NaCl. It should be noted that this unicellular organism even has a defending cell wall in contrast to viruses. Therefore, it is assumable that viruses and airborne germs are even more susceptible to salt impregnation [18]. This was verified in further studies. Paradigmatically, it was shown that alpha-coronavirus 1 infectivity was reduced by four orders of magnitude (99.99%) after contact of this virus with hypertonic saline-impregnated FFPs in pigs [11].

Additionally, contamination of sugar-containing media by airborne microorganisms was inhibited entirely at salt concentrations exceeding 5% NaCl [11]. As presented in these experiments, hypertonic salt solutions provoke a life-hostile impact on airborne microorganisms. Airborne pathogens did not contaminate the growth medium if salt concentrations exceeded 5% NaCl.

## 5. Strategies and Effects of Passive Immunization in Times of Pandemics

In contrast to the unspecific effects of salt denaturation, the reactions of organisms to virus infections are of particular interest. Antibodies against pathogens are critical for immune resistance in mammals, especially humans. After recovery, the immune system of such subjects produces far more immunoglobulins than needed. Consequently, immunoglobulins from such people may be harvested by isolation from the blood of former, i.e., convalesced, COVID-19 patients after recovery and contribute to the recovery of acutely infected patients. Unfortunately, IgG production decreases within months to years after the recovery of patients. Therefore, even very efficient antisera are only available for a limited period of time. Of course, immunization of mammals can be carried out to obtain immunoglobulin-producing cells, which could then be immortalized by fusion with, e.g., mouse myeloma cells [20]. Such immunoglobulins are so-called monoclonal antibodies (MCA) and are at least theoretically available in unlimited amounts.

Köhler and Milstein [21] received a Nobel award for this technique in 1984. One main disadvantage of MCA applied to humans in vivo is the possibility of anaphylactic reactions. The fusion of human B cells with mouse myeloma cells would result in human monoclonal antibodies (hMCA), which will not cause an anaphylactic reaction if applied to humans in vivo. Östberg and Pursch published such a technique in 1983 [22]. In addition, for conclusiveness, an hMCA against malondialdehyde-modified low-density lipoprotein was previously published [20].

## 6. Concept for Immortalization of COVID-19 Immunoglobulin-Producing Cells

From patients who had recovered from SARS-CoV-2 infections and who exhibited high titers of COVID-19 IgG, approximately 100 mL of peripheral blood could be drawn by arm vein puncture. After the separation of plasma, which simultaneously could be used to produce conventional IgG by, e.g., affinity chromatography, white blood cells (WBCs) could be separated from erythrocytes by the application of a Ficoll-Paque gradient. Isolated WBCs are then fused with mouse myeloma cells (e.g., SP-2) in an environment of polyethylene glycol (PEG) 400, which facilitates cell fusions. The reaction mixture is then transferred into 96-well plates in which a tissue culture medium containing hypoxanthine, aminopterin, and thymidine (HAT medium) is present. Only fused cells, which potentially can produce anti-COVID-19 IgG, can survive in that medium. After 24–48 h, the HAT medium is gradually removed from the cultures and replaced with a conventional medium enriched with 10% fetal bovine serum (FBS) or similar additives. Within a week, single clones of living cells appear and are then picked and transferred into 24-well plates after testing for the simultaneous presence of anti-COVID-19 IgG. Each clone is allowed to grow to confluence under regular testing, because many may lose their antibody production capability. The surviving clones with positive test results for anti-COVID-19 antibodies then become recloned by the following procedure: Isolated cells are seeded in 96-well plates at a concentration of 1 cell per well and again allowed to reach confluency, followed by new testing for hMCA directed against COIVD-19. Producing clones are then selected for immunoglobulin production in more significant amounts, which allows testing for the neutralizing properties of the anti-COVID-19 IgG. If these tests are satisfying, the clones can be used to start large-scale production, purification, and clinical trials for helpful therapeutic agents to reduce the consequences of SARS-CoV-2 infections. This is a technical description for producing such monoclonal antibodies as illustrated in Figure 1, among other techniques. For practical implementation, reference must be made to the ethical aspects and the clinical application, which requires detailed review and approval by the authorities.

## 7. Dry Blood Spot (DBS) Screening

A key criterion of DBS is the high level of concordance with serum [23] and plasma samples [24]. After successful initial immunization, there is a high correlation between SARS-CoV-2 antibody titers from DBS samples with a peak level and a decrease afterwards, depending on gender and age, which differs from unvaccinated subjects [23]. This humoral response was substantially decreased through a continuous decrease in IgG antibodies. This was more pronounced in males, as well as from the age of 65 years and in individuals with immunosuppression [25].

The advantage of DBS is that the sample poses minimal biohazard and is stable at ambient temperature. The antibody analysis does not have to be performed immediately; as such, a sample is insensitive to transport and environmental effects and can therefore be implemented economically to utilize an ELISA plate fully [26]. In addition to the determination of antibody titers using ELISA, it is also applicable for polymerase chain reaction (PCR) and micro assays [26,27]. It provides valuable information about vaccination success and offers the possibility of self-collection. Independent blood collection via the fingertip saves a visit to the doctor, which would be necessary for serum or plasma collection. This contact distance also significantly reduces transmission routes. Therefore, it dramatically assists the authorities in monitoring the immunity of the population, especially at times when distance is to be maintained.

Therefore, contactless monitoring of antibody titers using DBS is of great assistance in times of pandemics, both for the individual and the general population, and contributes to decision making by health authorities. As an example, the screening of the immune response of a male subject to COVID-19 vaccination, which was determined by the use of dried blood spots, is shown in Figure 2, as previously described [23].

## Figures and Tables

**Figure 1 viruses-15-01372-f001:**
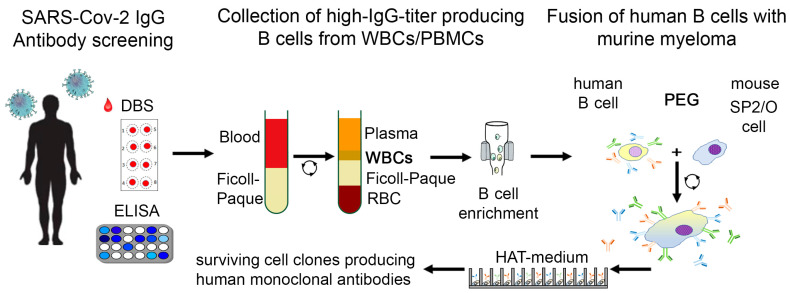
Conceptional workflow to isolate IgG-neutralizing antibodies from human donors and immortalization to obtain COVID-19 immunoglobin (IgG)-producing cells.

**Figure 2 viruses-15-01372-f002:**
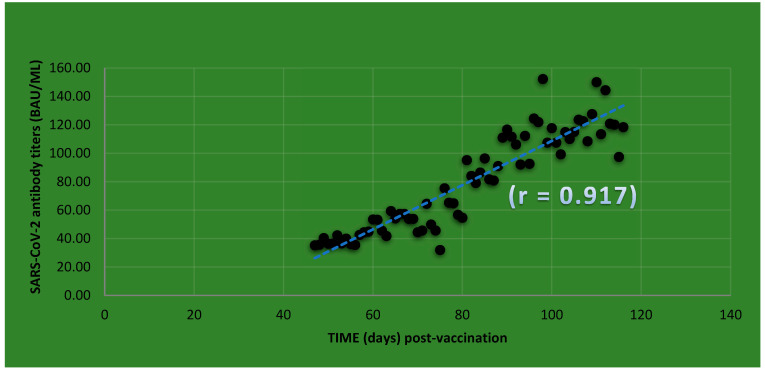
COVID-19 antibody titer screening of a single male subject (67 years) using dry blood spot samples after initial immunization with Vaxzevria^®^, adapted from [23].

**Table 1 viruses-15-01372-t001:** Antiviral activity of hypertonic saline coated on surgical masks, as previously described and adapted from [11]. Porcine alpha-coronavirus 1 (TGEV) was sprayed with an initial titer of 10^6^ TCID_50_/mL (50% tissue culture infectious dose per mL) on bare surgical masks with five pump-strokes from a spray flask (100 mL) at a distance of 20 cm. After 6 h of drying time, a square of 3 × 3 cm was cut from the central part of the mask, which was incubated for 16 h in 5 mL Dulbecco’s Modified Eagle’s Medium (DMEM) at room temperature. The eluent was used for the infection of a porcine testicular cell line. The titer after elution is indicated as mean, standard deviation (SD), minimum, maximum, and delta compared to the baseline titer. For the post-treatment effects of hypertonic saline, the experiment was repeated with a single modification; i.e., after the drying time, the masks were post-treated with hypertonic saline (10% NaCl) and dried for another 3 h. Regarding the pre-treatment effects of hypertonic saline, bare surgical masks were pre-coated with 10% sodium chloride 9 h before infection with TGEV. The remaining tasks were identical, as described above. Asterisks indicate a significant reduction in viral activity (*p* < 0.05) compared to bare surgical masks without hypertonic saline.

** *Titer* **		**Untreated**	**Post-Treated ***	**Pre-Treated ***
*Initial*	(TCID_50_/mL)	10^6.5^	10^6.5^	10^6.5^
*Mean*	(TCID_50_/mL)	10^5.86^	10^2.36^	10^1.74^
*SD*		10^1.05^	10^0.36^	10^0.18^
*Min*	(TCID_50_/mL)	10^5.12^	10^2.11^	10^1.62^
*Max*	(TCID_50_/mL)	10^6.61^	10^2.62^	10^1.87^

## Data Availability

Data are available on request from the corresponding author.

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
