# Peer review of "Thinking beyond Vaccination: Promising Add-On Strategies to Active Immunization and Vaccination in Pandemics—A Mini-Review"

_viruses, 2023, doi:10.3390/v15061372_

Round 1

Author Response

  1. R1: Limited scope: The authors primarily focus on two strategies, namely salt impregnation of filtering facepieces and passive immunization with antibodies. It does not provide an exhaustive overview of all potential add-on strategies that could be explored in pandemics.

A: We thank the Reviewer for this objection and agree that there are, of course, further protective measures which we did not address in this Brief Report. This mini-review described impregnating facepieces with hypertonic saline as an emergency measure because there are many errors during donning and doffing, as well as unconscious touching, which is the cause as even the majority of professional health workers did touch their masks with their hands (>93%). [Reszke et al. 2021, https://doi/10.3390/ijerph18020841]. The effectiveness of this measure against microorganisms has been confirmed several times, which is why this protective equipment is a remedy against careless handling, accidental contact and prolonged wearing. To emphasize this item, an explanatory sentence has been added at the end of the abstract: “The add-on measures were selected as examples for immediate, medium and long-term assistance and therefore did not claim to be exhaustive.”

As a medium-term measure, i.e., as soon as the technical prerequisites for the detection of antibodies are given, dried blood spot cards (DBS) offer a simple possibility to measure specific antibodies individually and in the entire population without contact, with a shallow risk of infection and easy transport, to gain information about the incidence of disease. Thus, the DBS method for diagnosing specific pandemic pathogens adds further possibilities in evading infectious hot spots like hospitals and ambulances.

A medium- to long-term measure is the production of human monoclonal antibodies. These could be of decisive benefit, especially for people at risk of allergies to vaccinations or for those whose immune system is suppressed - e.g., cancer or HIV patients. To provide more details of human monoclonal antibodies, it should be noted that mouse MCA may lead to anaphylactic reactions. Fusing human B-cells with mouse myeloma cells generates human MCA, which might significantly decrease the risk of this adverse reaction. This method was initially described by Östberg and Pursch in 1983 [Hybridoma 2(4):361-367], and a more recent publication confirmed this technique regarding a human monoclonal IgM antibody binding to MDA-LDL [Tatzber et al. 2017, https://doi.org/10.1155/2017/6047142]. It should be mentioned that this method is quite sophisticated compared to the original one, and its success is less frequent.

  1. R1: Lack of comprehensive evidence: This review mentions some experimental studies on the effects of salt solutions on microorganisms and viruses. However, the evidence presented is limited, and further research is needed to validate the effectiveness and safety of salt impregnation as a preventive measure.

A: We thank the reviewer for this hint and are congruent with the argument that efficacy for specific microorganisms must be adjusted by the corresponding hypertonic salt solution because there exist halophilic microorganisms that are probably not inactivated with a 10% salt solution. However, in our studies, we proved that the majority of airborne microorganisms, the yeast Saccharomyces cerevisiae (which is additionally equipped with a cell wall) and the coronavirus, Alpha-coronavirus 1, could be essentially killed or inactivated. The efficacy and safety of salt impregnation as a preventive measure or post-treatment for alpha-coronavirus 1 (a surrogate of SARS-CoV-2) was confirmed with an inactivation in the range of 104! We tried to present alternatives to vaccination in this Brief Report. We do not claim completeness, and it is not a new manuscript on the evidence of specific salt concentrations on the growth of specific microorganisms, which would exceed the scope of a Brief Report. We fully support any research activity in this regard.

  1. R1: Generalizability: The efficacy of salt impregnation of filtering facepieces may vary depending on the specific pathogens and environmental conditions. The review primarily discusses the potential impact on Alpha-coronavirus 1, which is a surrogate for SARS-CoV-2. The effectiveness against other viruses and microorganisms needs to be studied and confirmed.

A: Salt impregnation is a helpful general and unspecific emergency measure during an epidemic, pandemic, or other infectious diseases spread via aerosols. In the case of COVID-19, where no information was available regarding infectivity, mortality and so forth at the beginning of the pandemic, the effectivity of filtering facepiece coated with a hypertonic salt solution was proven in a surrogate virus (with a few to risk reduction for humans), i.e., Alpha-coronavirus 1, albeit comparable to SARS-CoV-2. Further investigations are inevitable for particular pathogens with corresponding environmental conditions - especially if the hypertonic salt solution (10% NaCl) shows no effect. In this regard, an explanatory sentence was added to page 5 - the last sentence. “An evaluation of the effective salt concentration is required for specific microorganisms, although a hypertonic salt solution of 10% sodium chloride was associated with significant inactivation in most microorganisms tested.”

  1. R1: Practical implementation: While the concept of salt impregnation seems straightforward, the review does not provide detailed guidance on the practical aspects of implementing this strategy. Factors such as the optimal salt concentration, impregnation methods, and durability of the impregnated facepieces require further investigation.

A: We thank the reviewer for the opportunity to discuss the practical implementation of coating filtering facepieces (FFP) with a hypertonic salt solution. In a recently published “Brief Report” (Tatzber et al. 2020, https://doi.org/10.3390/ijerph18147406), it is described in detail that the masks are sprayed with the hypertonic salt solution on the outside (if necessary, also on the inside). Alternatively, the mask can be dipped into the hypertonic saline solution, which is associated with a longer drying time. Additionally, we compared different impregnation methods, including ironing, but did not find striking differences from technique to process. The optimal salt concentration was found in this work to be 10% NaCl, although lower concentrations also showed high efficacy. Regarding durability, Rubino et al. 2020 (https://doi.org/10.1038/s41598-020-70623-9) even described increased effectiveness under humid conditions, as is commonly the case when the masks are worn for a more extended period. In another paper (Tatzber et al. 2021, https://doi.org/10.3390/ijerph18147406), the effect of hypertonic saline on FFP before and after contamination was elaborated. Although the preventive effect was slightly better, the post-treatment of the mask with hypertonic saline also provided similar results, i.e., a reduction in the viral activity 104. In this paper, all these relevant references are cited.

  1. R1: Potential risks and side effects: The authors briefly mention that salt impregnation may denature proteins and nucleic acids, leading to inactivation of viruses. However, it does not thoroughly address potential risks or side effects associated with this process. Further research is needed to assess the impact on the wearer's health, potential irritations, or respiratory issues caused by salt impregnation.

A: We thank the reviewer for this relevant question. We have addressed these issues in our publications and mentioned them accordingly. Just as every drug and every vaccination can have side effects, protective clothing could also have such problems, which we have pointed out and described in the paper by Tatzber et al. 2020, (https://doi.org/10.3390/ijerph18147406). If the inside is coated, this can be associated with a salty taste. In the case of hypersensitivity, e.g., in patients with high blood pressure or kidney problems, coating the inside only should be avoided. Furthermore, a high salt content can aggravate autoimmune diseases. However, there is no loss of pressure due to the coating. On the other hand, there are numerous indications of additional positive salt effects such as antimicrobial activity to support the immune system and fight pathogens, a reduction in inflammation, increased mucociliary elimination, dissolution of mucus, inhibition of bacterial growth, increase in phagocyte activity, and effects on general well-being and quality of life such as relaxation effects on the central nervous system (Tatzber et al. 2021, https://doi.org/10.3390/ijerph18147406). To point out any side effects of salt, we have inserted the following sentence (page 9 – last sentence): “However, salt impregnation should be avoided if hypersensitivity to salt is already present or occurs due to wearing an FFP coated with hypertonic saline.”

  1. R1: Ethical considerations: This review does not discuss ethical considerations related to passive immunization with antibodies from human sera. The use of monoclonal antibodies derived from immortalized B-cells raises ethical questions regarding the source of the antibodies and potential implications for donors.

A: As scientists, we have emphasized the technical possibility and implementation, whereby the ethical considerations concerning the material of human origin should be similar to those for regular blood donations. The legal provisions are the responsibility of the ethics committee or authorities. We have included this aspect on page 12 (last sentence) and referred to the ethical and legislative approvals. “This is a technical description for producing such human monoclonal antibodies, among other techniques. For the practical implementation, reference must be made to the ethical aspects and the clinical application, which requires detailed review and approval by the authorities.”

  1. R1: Regulatory approval and acceptance: The review does not address the regulatory approval process or public acceptance of these add-on strategies. Any new preventive measure or treatment would require rigorous evaluation, regulatory approval, and acceptance by the scientific community and the general public before widespread implementation.

We thank the reviewer for this aspect and fully agree that the approval and acceptance of such add-on strategies must be critically examined. Legal regulations are necessary, at least for using monoclonal antibodies in humans. In contrast, salt-coated FFPs have no contact with the human body that exceeds/overcomes the skin barrier. Therefore, the regulation should be included under personal protective devices. There are clear regulations for dry blood cards, which should also apply to viral diagnostics.

Reviewer 2 Report

This review has major flaws and I recommend to reject this review.

Here is a detailed explanation:

1) What is the relevance of killing viruses with salt? Viruses such SARS-CoV-2 are mainly transferred from person to person and do not survive long on the surface anyway. 

2) The section about antibodies is not updated-->There were thousands of papers about the hybridomna system. Why not talking about more advanced techniques such as single-B cell antibody cloning?

3) The rational of discussing DBS is not clear

4) Its seems that the authors combined 3 different topics into one review (decontamination with salt, antibodies and DBS).

Author Response

  1. R2: What is the relevance of killing viruses with salt? Viruses such SARS-CoV-2 are mainly transferred from person to person and do not survive long on the surface anyway. 

A: We thank the reviewer for the opportunity to answer this question in detail. In the specific case of alpha-coronavirus 1, i.e., a surrogate of SARS-CoV-2, it was shown that the viruses could largely survive on an untreated filtering facepiece. In contrast, virus activity was reduced 10,000-fold by preventive or subsequent coating with a hypertonic saline solution (Tatzber 2021, https://doi.org/10.3390/ijerph18147406). Polish scientists have found that only 6.8% of healthcare workers in Poland were compliant with the need to avoid touching the mask with their hands (Reszke et al. 2021, Int J Environ Res Public Health 18:841, https://doi.org/10.3390/ijerph18020841), which means there is a risk of infection, especially if the mask is worn for a long time. Thus, a hypertonic saline solution coating guarantees a relevant inactivation of viruses on FFPs.

  1. R2: The section about antibodies is not updated-->There were thousands of papers about the hybridomna system. Why not talking about more advanced techniques such as single-B cell antibody cloning?

A: The underlying intention was to demonstrate or remind that there is an alternative for groups of people who cannot produce antibodies per se, either by being allergic to vaccines or having a suppressed immune system. We focused on our know-how to produce human monoclonal antibodies to avoid possible anaphylactic reactions. We thank the reviewer for this hint but do not claim completeness regarding antibody production. We indicated in the text (page 12 – last sentence) that this is one of several possibilities to produce human monoclonal antibodies. “This is a technical description for producing such human monoclonal antibodies, among other techniques.”

  1. R2: The rational of discussing DBS is not clear

A: During the COVID-19 pandemic, rapid tests were used, which could have been more specific. As these dry blood cards can also be used for PCR tests, the health authorities' staff responsible for the home visits could be significantly relieved and, thus, resources saved. In addition, the level of immunization in the population is an essential criterion for decision-makers to make targeted social measures. Dry blood tests are inexpensive and can be sent by post at ambient temperature and collected at home. This relieves hot spots such as hospitals and outpatient clinics, and the samples can be collected in the laboratories and measured cost-effectively with full utilization of the ELISA.

  1. R2: Its seems that the authors combined 3 different topics into one review (decontamination with salt, antibodies and DBS).

A: It is correct that three measures are highlighted in this Brief Report, but on a singular topic already evident in the title: "Thinking beyond vaccination: promising add-on strategies to active immunization and vaccination in pandemics". This ranges from immediate measures such as coating of filtering facepieces with hypertonic saline to protect against infection due to improper donning and doffing. A medium-term possibility is obtaining information on the protection of individual immune reactions and antibody titres in the population as a whole, which would benefit those responsible for social measures. This is made possible by a collection system on dry blood cards, which brings additional advantages, i.e., a safe transport route without refrigeration and social distancing. Furthermore, finally, human monoclonal antibodies are an option for those individuals who have an allergic reaction to vaccines or those who cannot produce sufficient antibodies due to immunosuppressive therapy.

Reviewer 3 Report

This manuscript provides some useful strategies for protection against SARS-CoV-2. 

Presentation with tables for results of salt, passive immunization, and DBS screening would be beneficial. Using specific values or efficacy of these strategies is necessary.

I want to add one more comment for this manuscript. - What is the criteria of the reference selection and search strategies?

Author Response

  1. R3: Presentation with tables for results of salt, passive immunization, and DBS screening would be beneficial. Using specific values or efficacy of these strategies is necessary.

A: In addition to Figure 1 (which is already presented), we included a table regarding the efficacy of FFP coated with hypertonic saline on alpha-coronavirus 1, as well as a second figure monitoring an individual's antibody titre after coronavirus vaccination, which was analyzed using the dry blood spots. See pages 6-7 (Table 1) and 14 (Figure 2) for details.

  1. R3: I want to add one more comment for this manuscript.- What is the criteria of the reference selection and search strategies?

A: This Brief Report describes immediate, medium and long-term alternative approaches that could benefit humanity in the wake of a pandemic. During the COVID-19 pandemic, most research projects were limited or wholly suspended. During this phase, our research group has been thinking about ways to help the general public and has conducted studies during the lockdown to generate data and evidence for our hypotheses. This Brief Report is intended to draw attention to alternative options that can be taken beyond immunization with vaccines to protect against unknown (or known) sources of the danger of infection. In this respect, both the search strategy and the references were selected. It is worth mentioning that as early as April 2020, two research groups on entirely different continents, i.e., Canada (Rubino et al. Sci Rep 2020 10:13875; https://doi.org/10.1038/s41598-020-70623-9) and Austria (Europe) (Tatzber et al. Prev Med Rep 2020, 20, 101270; https://doi.org/10.1016/j.pmedr.2020.101270) were considering the germicidal effect of a hypertonic saline solution.

Round 2

Reviewer 2 Report

The review can now be published

The review can now be published